# Long Chain N3-PUFA Decreases ACE2 Protein Levels and Prevents SARS-CoV-2 Cell Entry

**DOI:** 10.3390/ijms232213825

**Published:** 2022-11-10

**Authors:** Shiqi Huang, Carla G. Taylor, Peter Zahradka

**Affiliations:** 1Department of Food and Human Nutritional Sciences, University of Manitoba, Winnipeg, MB R3T 2N2, Canada; 2Canadian Centre for Agri-Food Research in Health and Medicine, St. Boniface Hospital Albrechtsen Research Centre, Winnipeg, MB R2H 2A6, Canada; 3Department of Physiology and Pathophysiology, University of Manitoba, Winnipeg, MB R3E 0W2, Canada

**Keywords:** ACE2, ACE1, omega-3 fatty acids, long COVID, SARS-CoV-2

## Abstract

Angiotensin-converting enzyme 2 (ACE2) is a target of interest for both COVID-19 and cardiovascular disease management. Even though lower ACE2 levels may be beneficial in SARS-CoV-2 infectivity, maintaining the ACE1/ACE2 balance is also crucial for cardiovascular health. So far, reports describing conditions capable of altering ACE2 protein levels, especially via dietary components, are limited. In this study, the effects of omega-3 polyunsaturated fatty acids (n3-PUFA) on the protein levels of ACE1 and ACE2 in rodent tissues, human endothelial and kidney cell lines, and human plasma were examined. The ability of n3-PUFA to affect the entry of the SARS-CoV-2 pseudovirus into cells was also tested. Docosahexaenoic acid (DHA), and in some cases eicosapentaenoic acid (EPA), but not α-linoleic acid (ALA), reduced both ACE1 and ACE2 (non-glycosylated p100 and glycosylated p130 forms) in the heart, aorta, and kidneys of obese rats, as well as in human EA.hy926 endothelial and HEK293 kidney cells. Dietary supplementation with either DHA or ALA had no effect on plasma soluble ACE2 levels in humans. However, treatment of HEK293 cells with 80 and 125 µM DHA for 16 h inhibited the entry of the SARS-CoV-2 pseudovirus. These results strongly suggest that DHA treatment may reduce the ability of SARS-CoV-2 to infect cells via a mechanism involving a decrease in the absolute level of ACE2 protein as well as its glycosylation. Our findings warrant further evaluation of long-chain n3-PUFA supplements as a novel option for restricting SARS-CoV-2 infectivity in the general population.

## 1. Introduction

The current COVID-19 pandemic, caused by the SARS-CoV-2 virus, has claimed millions of lives [1]. Besides being a respiratory virus, SARS-CoV-2 is increasingly recognized as a vascular pathogen causing severe cardiovascular complications [2] and renal dysfunction [3,4]. The mortality and severity of COVID-19 is significantly higher in patients with pre-existing health conditions, such as cardiovascular disease (CVD), hypertension, and diabetes [5]. Endothelial dysfunction is one of the common underlying phenomena in diseases that elevate COVID-19-risk, such as CVD [6,7]. SARS-CoV-2 infection can contribute to endothelial dysfunction by triggering inflammatory and immune responses that cause a cytokine storm, hypoxia, and elevated oxidative stress [6,7]. Cytokine storms and associated endothelitis (inflammation of the endothelium) may be critical factors in the processes that promote organ failure, and subsequent COVID-19 morbidity and mortality [8,9]. Consequently, identifying novel therapies that can stabilize endothelial function may provide alternative strategies to ameliorate the poor clinical outcomes associated with COVID-19 infection.

Angiotensin-converting enzyme 2 (ACE2) is the receptor used by SARS-CoV-1 and SARS-CoV-2 to enter human cells. Profiling ACE2 protein levels has shown that ACE2 is expressed across a broad spectrum of cell types, including endothelial cells, cardiomyocytes, and renal proximal tubule cells, all of which exhibit higher levels of ACE2 than the pulmonary system [10,11]. Though this information is useful, it does not provide information regarding conditions that may increase or decrease ACE2 protein levels. For instance, there is evidence that cytokines associated with chronic disease can elevate ACE2 levels in pulmonary artery endothelial cells [12], but little is known regarding its regulation in other tissues.

Both the presence of ACE2 in endothelial cell membranes and its catalytic activity are reduced by the shedding of ACE2 in response to the cellular entry of SARS-CoV viruses [11]. Based on these data, as well as evidence that ACE1 levels are unchanged by infection [13], it is likely that SARS-CoV-2 infection causes an imbalance in the relative levels of angiotensin-converting enzyme 1 (ACE1) and 2 (ACE2). ACE1 converts angiotensin I to angiotensin II (AngII), a potent vasoconstrictor, whereas ACE2 hydrolyses AngII to angiotensin-(1–7), a vasodilator [14]. An imbalance in ACE1/ACE2 caused by infection would result in enhanced AngII production, promoting the progression of thrombotic and inflammatory processes and increased vasoconstriction [14]. When applied to the circulatory system, it has been suggested that this imbalance plays a major role in the injury caused by SARS-CoV-2 in different organ systems, due to increased AngII signaling [14]. Based on this perspective, ACE1 inhibition and angiotensin receptor blockers would be beneficial for COVID-19 patients [14], as has been reported for SARS-CoV [13].

Modulation of ACE2 expression has been proposed to assist in ameliorating the negative effects of SARS-CoV-2 infection [15]. In addition to various pharmacological agents, many natural products have been examined for their ability to affect ACE2 [16]. At the same time, investigations into whether induced ACE2 changes also affect ACE1, thereby altering the ACE1/ACE2 balance, or whether the bioactive agent actually alters SARS-CoV-2 entry into cells and/or its cellular actions, typically have not been examined. The same can be said for dietary fatty acids, which we have shown are able to alter the function of endothelial cells [17,18]. Although it has been suggested that omega-3 polyunsaturated fatty acids (n3-PUFAs) may be beneficial in SARS-CoV-2 management, especially the long chain n3-PUFAs (LCn3-PUFA) with ≥20 carbon atoms such as docosahexaenoic acid (DHA) and eicosapentaenoic acid (EPA) [19,20], the ability of these molecules to modulate SARS-CoV-2 infectivity has not been examined. To date, research examining the effect of PUFAs on *ACE2* mRNA levels in adipocytes and renal tissue has yielded contradictory results [21,22]. Given the positive actions of PUFA on endothelial cells [23], and the broad utilization of n3-PUFAs in supplement form by the general population, an investigation of the effect of several n3-PUFAs on ACE2 protein levels was undertaken. During this study, the effects of n3-PUFAs on ACE1 and ACE2 protein levels in rodent tissues and human cell lines were examined, as well as the ability of n3-PUFAs to block SARS-CoV-2 pseudovirus entry into cells. Additionally, we assessed the ability of n3-PUFA supplementation to modulate circulating ACE2 levels in humans.

## 2. Results

### 2.1. DHA Modulates Tissue Levels of ACE2 and ACE1 in Rats

The effects of dietary PUFA on the SARS-CoV-2 cellular receptor, ACE2, was explored by Western blotting of various tissue lysates prepared from genetically obese *fa/fa* Zucker rats that had been placed on diets enriched in either an n3-PUFA (α-linoleic acid (ALA), EPA, or DHA), or n6-PUFA (linoleic acid, LA) for 8 weeks (Figure 1a–d). The tissues investigated were of the heart (Figure 1a), aorta (Figure 1b), kidney (Figure 1c), and lung (Figure 1d), which are known to be key targets affected by SARS-CoV-2 infection [11]. Quantification of the blots (Figure 1e) revealed that the DHA diet significantly reduced ACE2 protein levels compared with ALA and/or EPA diet groups in rat heart, aorta, and kidney, but not lung; this result was obtained for both the major 130 kDa band (heart and kidney), as well as the minor 100 kDa band (heart and aorta). The p130 band was established as a glycosylated form of the p100 band by treatment with peptide-*N*-glycosidase F (PNGase F), which shifted all of the p130 form to the p100 form (Figure 1f). These data are significant because ACE2 glycosylation plays a critical role in viral entry into the cell [24,25]. Thus, DHA treatment is able to reduce the levels of the major form of ACE2, which are required for infection.

As the balance of ACE1 and ACE2 levels is important for many aspects of vascular health due to the impact of these enzymes on AngII formation and degradation, respectively [14], we also examined the effect of the PUFA diets on ACE1 in the rat heart, aorta, and lung. It was observed that ACE1 levels did not change in the lung, whereas ACE1 was lower in the rat heart and aorta in DHA-fed rats compared with the LA and/or ALA groups (Figure 1g), which paralleled the DHA-mediated reductions in ACE2, as shown in Figure 1e. Therefore, no change in the ACE1/ACE2 ratio was observed in the rat heart and aorta (Figure 1h). These results suggested that though ACE2 levels were reduced by DHA, an imbalance in the renin-angiotensin system was not likely to occur.

### 2.2. DHA Reduces ACE2 Levels of Human Endothelial Cells

Modulation of endothelial cell phenotypes occurs in response to injury, allowing cells to migrate and proliferate, which is not typical in normal healthy vasculature [18]. Thus, endothelial cells in different states, activated and proliferative versus healthy and quiescent, may respond differently to the same stimulus. To explore the contribution of these different conditions on the ability of n3-PUFA to modulate ACE2 and ACE1, human EA.hy926 endothelial cells were used as they can be placed into both proliferating (activated) and quiescent (inactive) states [26]. The presence of ACE1 and ACE2 (Figure 2a–d) was quantified 8 and 24 h after treatment (Figure 2e,f).

In growing (activated) cells, both p130 and p100 ACE2 levels were significantly lower in response to DHA treatment at both 8 (all DHA concentrations) and 24 h (80 and 125 µM DHA), relative to vehicle control (Figure 2e upper left and right panels). Though no change was observed with ALA treatment, EPA caused a reduction in ACE2 as well, but it required a considerably higher concentration (125 µM) to produce an effect at 8 h comparable to that of the 20 µM DHA for both the 130 and 100 kDa forms. With a longer treatment time (24 h), however, the ACE2-lowering effects of DHA and EPA were similar. In contrast to the results obtained in growing cells, n3-PUFA treatment with DHA (except 40 µM) and 20 µM EPA for 8 h increased the levels of both p130 and p100 ACE2 in quiescent cells (Figure 2e left panel); however, this increase was not observed at 24 h (Figure 2e lower right panel), suggesting that the effects of the PUFA treatments are transient. Furthermore, 24 h treatment with 125 µM EPA led to a reduction in p130 ACE2 relative to the vehicle control, whereas the p100 form was not detected after 24 h of treatment time.

Similar to the results in rat tissues, n3-PUFA treatment of EA.hy926 cells also affected ACE1 levels (Figure 2f). In growing cells, 20 µM DHA increased ACE1 levels relative to the vehicle control at 8 h, whereas 20 to 80 µM DHA decreased ACE1 levels relative to vehicle control at 24 h. Apart from reduced ACE1 levels by 80 and 125 µM EPA compared with vehicle control at 24 h, no other effects of EPA or DHA at 8 or 24 h were observed in growing cells. All concentrations of EPA were highly potent for reducing ACE1 levels after 8 h when added to quiescent cells, but not at 24 h. In quiescent cells, 80 µM DHA significantly lowered ACE1 at 8 h, but 80 and 125 µM DHA significantly increased ACE1 after 24 h compared with vehicle control. In addition, 125 µM ALA decreased ACE1 levels at 8 h and increased ACE1 levels at 24 h compared with the vehicle control in quiescent cells.

In summary, DHA and EPA could reduce ACE1 protein levels in both growing and quiescent EA.hy926 cells, but the LCn3-PUFAs only decreased ACE2 levels in growing cells and not in quiescent ones. These results indicate that the effects of n3-PUFAs on EA.hy926 cells were rather complex, and were dependent on treatment time, concentration, and cell growth state.

### 2.3. DHA Supplementation Does Not Alter Circulating ACE2 Levels

Although ACE2 is typically membrane-bound, this protein can be shed from endothelial cells by the actions of several proteases, which release catalytically active ACE2 into the circulation [27]. Circulating ACE2 (or plasma ACE2, known as sACE2) has been proposed as a biomarker for CVD risk [28]. SARS-CoV-2 infection was also found to elevate sACE2 [29], and higher sACE2 levels and activity are associated with increased COVID-19 severity [30,31], as sACE2 can also mediate viral cell entry [27]. Therefore, to determine whether dietary n3-PUFA affects sACE2 levels, sACE2 was quantified by ELISA in the plasma of 34 subjects before and after a 4-week supplementation with ALA (Figure 3a) or DHA (Figure 3b). Overall, 29 sets of results were within the detection range of the kit. The average intraplate variation was 5.97%, whereas the inter-plate variation was 8.00%. sACE2 levels were unchanged by supplementation with either of these n-3 PUFAs (*p* = 0.405 for ALA and *p* = 0.305 for DHA, Figure 3a,b). Subgroup analysis showed no significant difference before or after supplementation in either the obese subgroup (15.29 ± 4.48 vs. 15.68 ± 4.66, before and after ALA supplementation, respectively; 14.33 ± 4.18 vs. 14.98 ± 4.25, before and after DHA supplementation, respectively) or the healthy subgroup (4.43 ± 1.55 vs. 4.60 ± 1.52, before and after ALA supplementation, respectively; 3.33 ± 1.16 vs. 4.41 ± 1.47, before and after DHA supplementation, respectively). Furthermore, the response among different individual subjects was diverse, with sACE2 levels going up after ALA supplementation but coming down after DHA for some, whereas the opposite occurred in other subjects. The study used a randomized cross-over study design, with a minimum 4-to-6-week washout between supplementation phases [32,33], and a non-significant paired *t*-test between the two day 0 sample sets confirmed that there were no potential carry-over effects of PUFA supplementation. Although the average sACE2 levels of healthy subjects seemed lower than that of obese subjects, we did not find the two groups to be statistically different (Figure 3c, *p* = 0.109). Subgroup analysis for sex differences revealed that healthy male participants had higher sACE2 than healthy females (Figure 3d), but there was no significant effect of supplementation for either males or females.

Further correlation analysis of the data revealed that sACE2 levels were associated with certain health parameters (Table 1), such as body mass index (BMI), blood pressure (BP), high-density lipoprotein-cholesterol (HDL-C), and glucose (Glc) levels. As shown in Table 1, baseline sACE2 levels were positively associated with systolic BP (SBP) and fasting glucose levels, while negatively associated with HDL-C. Further subgroup analysis revealed that in obese participants, there were trends for negative correlations between sACE2 levels and total cholesterol (TC) and HDL-C, as well as a positive trend with glucose levels. In healthy participants, SBP and HDL-C were positively correlated with sACE2 levels (glucose levels were not measured in this study).

Overall, n3-PUFA supplementation was not found to affect sACE2 levels in human subjects. Even so, sACE2 was correlated with some health parameters, such as BP, cholesterol levels, and glucose levels. In addition, sex differences were observed for sACE2 levels.

### 2.4. DHA Treatment Reduces Pseudovirus Entry into Cells

The kidney is highly susceptible to SARS-COV-2 infection, which can lead to a variety of pathological conditions [3,34]. As seen with other cells, SARS-CoV-2 infection requires expression of ACE2, and this protein is found on the surface of proximal tubule cells and podocytes [4]. In concordance with the results of Figure 1, showing that ACE2 is present on the rat kidney, ACE2 is also expressed by the human embryonic kidney cell line, HEK293 (Appendix A). As treatment with DHA resulted in a decline in ACE2 (both non-glycosylated p100 and glycosylated p130 forms) on HEK293 cells (Appendix A), these cells were an ideal model for investigating the effects of n3-PUFA treatment on their susceptibility to SARS-CoV-2 infection. Based on the data shown in Appendix A, concentrations of DHA up to 125 µM were used to treat the cells for 16 h before transduction for 24 h with a recombinant SARS-CoV-2 spike protein/GFP hybrid protein. This construct made it possible to visualize entry of the GFP-tagged protein into cells and quantify entry based on fluorescence intensity. Visual monitoring of the treated cells clearly illustrated that high concentrations of DHA, but not ALA, decreased entry of the recombinant protein into the target cells expressing ACE2 (Figure 4a). Although some toxicity was observed when the cells were treated with 125 µM DHA, based on the reduction in Hoeschst staining that was consistent with previous reports [26,35], the normalized fluorescence intensity readings from the same batches of cells (Figure 4b) indicated that viral infection of the cells was reduced after treatment with 80 and 125 µM DHA compared with the 20 µM DHA treatment (*p* = 0.029 and 0.012, respectively). Overall, these results showed that DHA was not only able to reduce cellular ACE2 levels, but also reduce viral entry into the cells, suggesting that DHA treatment may plausibly be employed as a therapeutic to lower the susceptibility of cells to SARS-CoV-2 by decreasing the number of ACE2 molecules available as entry points to the cells.

## 3. Discussion

This study is the first to report that the LCn3-PUFAs, particularly DHA, can reduce the protein levels of ACE2, the cellular receptor of SARS-CoV-2, both in vivo (heart, aorta, and kidney of obese rats) and in vitro (human EA.hy926 endothelial and HEK293 embryonic kidney cells). Both glycosylated and non-glycosylated forms of ACE2 were reduced by DHA. This decrease in the ACE2 of rodent tissues was accompanied by a parallel decline in ACE1 protein, thus maintaining the balance of ACE1 to ACE2, in vivo. Additionally, the entry of a pseudovirus expressing SARS-CoV-2 spike proteins into live cells was hindered in response to DHA treatment conditions that lowered ACE2 protein levels. This model system, which provided an idealized approach to monitor viral entry into cells, strongly suggests that DHA treatment may reduce the ability of SARS-CoV-2 to infect cells, via a mechanism involving a decrease in the absolute level of ACE2 protein as well as its glycosylation. This novel mechanism of action of LCn3-PUFA (Figure 5) suggests a novel approach for restricting SARS-CoV-2 infection and potentially limiting the adverse cardiovascular outcomes associated with COVID-19.

The ability to reduce ACE2 levels with LCn3-PUFA treatment, as reported in this study (Figure 1, Figure 2 and Appendix A), may provide a new paradigm for fundamentally decreasing the effectiveness of SARS-CoV-2 to infect the host. The advantage of targeting ACE2 levels is highlighted by the fact that the protective immunity gained from vaccination or previous SARS-CoV-2 infection is short term, and the phenotypic instability of the virus contributes to the high frequency of new variants that can evade vaccine- and/or infection-derived immunity [36]. An alternative approach that does not yet depend on immunity and capable of lowering infectivity would be an ideal adjunct preventative strategy. As suppressing ACE2 has been shown to inhibit SARS-CoV-2 infection in vitro, even for the Omicron and Delta variants [37,38], the downregulation of ACE2 is a very promising therapeutic strategy. As opposed to antibody-based methods to reduce ACE2 levels [38], a readily available oral supplement such as LCn3-PUFA provides a novel option for further evaluation as a means to lower the risk of SARS-CoV-2 infection in the general population.

Similar to our results showing that LCn3-PUFAs reduced ACE2 protein levels in both animal tissues (Figure 1e) and cell culture (Figure 2e and Appendix A), it has been reported that DHA and its metabolites were able to reduce *Ace2* mRNA in murine kidney [39] and porcine adipocytes [21]. Therefore, DHA and EPA may lower ACE2 levels by reducing *ACE2* gene transcription. Tseng et al. [21] also found that the reduction of *Ace2* mRNA by 50 µM DHA (but not EPA) possibly occurred via pathways for eicosanoid production, suggesting that DHA and/or its derivatives were more potent than EPA. This is also consistent with our finding that DHA treatment was more efficient in reducing ACE2 than EPA (Figure 1e, Figure 2e and Appendix A). Additional efficacy may arise through a reduction in the glycosylated form of ACE2 by LCn3-PUFAs, as shown for the first time in this study, as glycosylation is crucial in mediating SARS-CoV-2 cell entry via ACE2 [24]. ACE2 can be extensively glycosylated [40], as indicated by the predominance of the p130 band in animal tissues (Figure 1a–d) and cultured cells (Figure 2a–d and Appendix A), and its six *N*-glycosylation sites are occupied with diverse glycans, each having a divergent effect in the context of SARS-CoV-2 infection. For example, glycosylation at N90 can hinder spike-ACE2 binding, whereas glycosylation at N322 strengthens it [25]. With only Western blotting data in this study, we cannot speculate on whether changes in glycosylation occurred at specific sites, or whether the glycan content varied between tissues [40]. However, alterations in p130 levels could have resulted from a decrease in ACE2 expression, which is suggested by the concomitant disappearance of the p100 band. Consequently, we infer that n3-PUFA likely affects both the glycosylation of ACE2 and its expression. These results are consistent with the only publication to date to report that n3-PUFA consumption modulated the glycosylation of plasma proteins [41], and the present findings extend this observation to the cell surface proteins of a variety of tissues and cell types.

The present study also demonstrated that DHA could block entry of a pseudovirus expressing SARS-CoV-2 spike protein from entering cells (Figure 4), likely by a mechanism involving the lowering of both non-glycosylated and glycosylated forms of ACE2. LCn3-PUFA, especially DHA, is believed to interfere with viral infection, not limited to SARS-CoV-2 infection, by preventing viral replication processes and ACE2-mediated viral entry via altered lipid raft formation [42]. Here, we provided another possible interference route, by reducing ACE2 protein levels. Although Goc et al. [43] recently reported that ALA and EPA could reduce the cell entry of a fluorescence-tagged pseudovirus into cells expressing recombinant ACE2 proteins, they attributed this effect to a PUFA-dependent inhibition of spike protein binding to ACE2 in their system. Furthermore, their system was not designed to examine the effect of PUFA treatment on native ACE2 levels. As well, the high concentrations of DHA they required to reduce spike protein-ACE2 binding were at least two times greater than the highest concentration employed in our study. A cross-sectional study revealed that human plasma DHA concentrations can vary from 7.2 to 237.5 µM with 90% of samples between 47.8 to 138.0 µM [44]. Therefore, the DHA concentrations (20–125 µM) used in this study are well within the physiological range of DHA levels that can be reached in the general population through supplementation. A more recent paper also found that DHA and EPA within this concentration range reduced SARS-CoV-2 pseudovirus cell entry, as well as ACE2 protein levels that were elevated by trimethylamine-N-oxide treatment of human endothelial progenitor cells [45]. They further revealed that the reduction in both pseudovirus infection and ACE2 expression was mediated by NF-κB signaling, which could explain our finding that DHA treatment elicited a decline in ACE2 levels in the heart, aorta, and kidney, but not the lung. Specifically, there have been several reports indicating DHA suppresses NF-κB signaling in these three tissues [46,47,48], whereas such a mechanism has not been observed in the lung.

Our results also showed, for the first time, that LCn3-PUFAs decreased ACE1 protein levels (Figure 1g, Figure 2f and Appendix A). This reduction in ACE1 could be one of the reasons that blood pressure was lowered with n3-PUFA treatment [49], thus explaining their utility in hypertension management [50]. The concomitant decrease in ACE1 and ACE2 levels by LCn3-PUFAs could prevent the potentially adverse effects of ACE2 reduction, which happens in response to SARS-CoV-2 infection on the cardiovascular system by promoting AngII accumulation [14]. Thus, an added advantage of n3-PUFA treatments is that they do not adversely affect the balance of ACE1/ACE2 in vivo (Figure 1h).

Higher sACE2 is found in both COVID-19 [30,31] and CVD [28,51] patients than in the healthy population, but n3-PUFA supplementation did not affect sACE2 levels in either obese or healthy subjects (Figure 3), possibly due to the limited 4-week duration of the intervention. On the other hand, sACE2 levels were correlated with BP, TC, HDL-C, and glucose levels in individuals without CVD (Table 1). Similar associations have been previously reported in CVD [51], whereas some correlations, such as sACE2 levels and BP, hold true even in COVID-19 patients [52]. The individual variability in sACE2 may be explained in part by the influence of genetics on sACE2 levels [51]. Additionally, males had higher sACE2 levels (Figure 3d), as reported previously [30,51], which may be one reason why males are more susceptible to COVID-19-related morbidity and mortality [5]. Obesity is associated with more severe COVID-19 outcomes [5], and sACE2 was found to be higher in obese individuals [51]. Although we did not find statistically significant differences, most likely due to our limited sample size, the average sACE2 levels were approximately three times higher in the obese subgroup compared with the healthy subgroup (Figure 3c). Elevated sACE2 may thus serve as a bridge for the vicious cycle between COVID-19 and CVD.

In vivo, LCn3-PUFA decreased ACE2 in tissues of obese *fa/fa* Zucker rats (Figure 1e), which have endothelial dysfunction [53], whereas in vitro, this reduction was only observed in the growing, activated state of EA.hy926 cells (Figure 2e), which likewise represented endothelial dysfunction [26]. There is evidence to suggest that endothelial dysfunction underlies long COVID [2,6,54], the persistence of COVID-19-related multi-system symptoms after recovery from the initial infection, which affects more than 30% of COVID-19 patients worldwide [1,55]. LCn3-PUFA, therefore, may be beneficial in managing long COVID as well. Furthermore, this growth-state-dependent effect of LCn3-PUFA (Figure 2) may be due to the different receptors and/or signaling pathways that interact with PUFA in the two growth states as DHA is also a signaling molecule [56]. In addition to growth state, the concentration-dependent effects of LCn3-PUFA on both ACE1 and ACE2 levels in cells and SARS-CoV-2 pseudovirus cell entry need to be considered. Specifically, appropriate therapeutic dosing of DHA and EPA (or fish oil) should be investigated to determine the optimal protective conditions for both viral infection and long COVID. Furthermore, the possible chronic effects of low dose supplementation in vivo compared with a higher dose treatment regimen should not be underestimated.

## 4. Materials and Methods

### 4.1. Animal Tissue Protein Extraction

Rat heart, aorta, kidney, and lung tissues were obtained from a previous diet intervention study [57]. Briefly, 6-week-old male obese *fa/fa* Zucker rats were randomized into four PUFA diet groups: ALA, EPA, DHA, and LA. Animals remained on the diets for 8 weeks. Tissues were flash frozen in liquid nitrogen and stored at −80 °C until protein lysates were prepared by homogenization in 3× sample buffer (3× SB; 1× SB = 62.5 mM Tris-HCl, pH 6.8, 1% sodium dodecyl sulphate (SDS), 10% glycerol); the protein lysates were stored at −80 °C.

### 4.2. Cell Culture and Treatments

EA.hy926 cells (Catalog #: CRL-2922, American Type Culture Collection, Manassas, VA, USA) were cultured on 12-well plates to sub-confluent and quiescent states on Matrigel^®^ (Catalog #: 356231, Corning^®^, Corning, NY, USA), as previously described [26]. HEK293 cells (Catalog #: CRL-1573, American Type Culture Collection, Manassas, VA, USA) were cultured in Dulbecco’s modified Eagle medium (DMEM, made from DMEM powder (Catalog #: 12800-82, Gibco, Thermo Fisher Scientific, Inc., Waltham, MA, USA), supplemented with 20 mM HEPES (Catalog #: 391340, Merck Millipore, Burlington, MA, USA), 100 units/mL penicillin/streptomycin (Catalog #: 15140, Gibco, Thermo Fisher Scientific, Inc., Waltham, MA, USA), and 10% fetal bovine serum (Catalog #: SH30071, Hyclone, Logan, UT, USA). Cells were seeded on 12-well plates at 2 × 10^4^ cells/cm^2^ two days before treatment, with various concentrations of ALA, EPA, and DHA (Catalog #: 90210, 90110, and 90310, respectively, Cayman Chemical, Ann Arbor, MI, USA). After 8 or 24 h (as indicated in the figures), cells were lysed with 100 µL 2× SB. The fatty acids were initially dissolved in ethanol before conjugation to fatty acid-free bovine serum albumin (BSA; Catalog #: 10775835001, Roche, Basel, Switzerland) in phosphate-buffered saline (PBS) for delivery to the cells [26]. An equivalent amount of ethanol in 5% BSA-PBS was used as the vehicle control.

### 4.3. Sample Preparation and Western Blotting

Tissue and cell protein lysates were subjected to a brief sonication and centrifugation prior to protein quantification with a BCA assay kit (Catalog #: 23225, Pierce, Thermo Fisher Scientific, Inc., Waltham, MA, USA). Western blotting was done as previously described [26]. Briefly, protein (15 µg per lane) was separated by SDS-PAGE prior to transfer to PVDF membrane. Rabbit anti-ACE2 polyclonal primary antibody (Catalog #: 3217, ProSci, San Diego, CA, USA) and StarBright Blue 700 Goat Anti-Rabbit secondary antibody (Catalog#: 12004162, Bio-Rad, Hercules, CA, USA) were used to detect ACE2. Mouse anti-ACE1 monoclonal primary antibody (Catalog #: sc-23908, Santa Cruz, Dallas, TX, USA), together with an HRP-conjugated goat anti-mouse secondary antibody (Catalog #: 1706516, Bio-Rad, Hercules, CA, USA) were used to detect ACE1. Ponceau S staining (Catalog #: 97063-650, VWR, Radnor, PA, USA) was used to assess total protein loading for normalization [58]. Blot images were captured with a ChemiDoc Imager (Bio-Rad, Hercules, CA, USA) and quantified using Image Lab software version 6.0.1 (Bio-Rad, Hercules, CA, USA).

### 4.4. Sample De-Glycosylation

To confirm *N*-glycosylation of ACE2, treatment with PNGase F (Catalog #: P7367, Sigma-Aldrich, St. Louis, MO, USA), followed by Western blotting was employed. A 25 µL aliquot of post-centrifugation lysate was mixed with 20 µL of 50 mM sodium phosphate, pH 7.5, and heated to 100 °C for 10 min prior to the addition of 5 µL of 15% Triton^®^ X-100. Then, 10 µL of 500 units/mL PNGase F was added to the mixture, followed by incubation at 37 °C for 2 h, at which point the reaction was stopped by heating the mixture to 100 °C for 5 min. The protein content of both PNGase F-treated and non-treated aliquots was then quantified before monitoring the apparent molecular mass by Western blotting.

### 4.5. Circulating ACE2 ELISA

Frozen plasma samples were obtained from two previous randomized double-blind crossover studies [32,33]. The samples were from 34 subjects (healthy lean men and women [NCT02317588] and obese women [NCT03583281]), supplemented with ~4 g/d of ALA or DHA daily for 28 days. sACE2 levels, before and after each supplementation phase, were measured in duplicate with the Human ACE-2 DuoSet^®^ ELISA kit (Catalog #: Dy933, R&D Systems, Minneapolis, MN, USA), according to the manufacturer’s protocol. Where applicable, samples were diluted with 1× Reagent Diluent. Plates were read on a FLUOstar Omega plate reader (BMG LABTECH, Ortenberg, Germany) at 450 nm, with 570 nm used for wavelength correction.

### 4.6. Pseudo SARS-CoV-2 Cell Entry Assay

To assess whether n3-PUFA could affect viral entry into cells, we utilized a pseudovirus system based on BacMam technology that expressed fluorescent-gene-fused SARS-CoV-2 spike protein (Catalog #: C1110G, Montana Molecular, Bozeman, MT, USA) and HEK293 cells. In this model, the baculoviruses are pseudotyped with spike protein that can recognize and interact with the ACE2 expressed by the HEK293 cells. When the spike protein binds to ACE2, pseudovirus entry into the cell delivers a fluorescent reporter into the host cell nucleus that is detected via its green fluorescence. Higher fluorescence intensity indicates greater spike–ACE2 interaction and higher levels of pseudovirus entry. Briefly, the cells were cultured and seeded, as described above, into black-walled, clear-bottom 96-well microplates. Two days after seeding, the cells were treated with the indicated concentrations of ALA or DHA for 16 h. Then, cells were incubated for an additional 24 h with the SARS-CoV-2 pseudovirus, according to the manufacturer’s transduction protocol, after which cellular entry of the pseudovirus was assessed via fluorescent intensity. Specifically, the transduction mix (per well) was comprised of 5 µL SARS-CoV-2 pseudovirus, 0.6 µL sodium butyrate, and culture media to achieve a total volume of 50 µL. For non-transduction controls, sodium butyrate and media were added to the respective PUFA-treated cells. After incubation in the dark, the cells were stained with Hoechst 33342 for 10 min at 37 °C, and then the media was replaced with 1× PBS for fluorescence reading with the FLUOstar Omega plate reader (BMG LABTECH, Ortenberg, Germany) and imaging with a fluorescent microscope (Olympus, Tokyo, Japan). Three wells were prepared for each treatment condition as technical replicates for 3 independent cell passages. Results were excluded from the analysis if they were from wells with blue fluorescence readings lower than that of the non-Hoechst-stained well and/or the normalized fluorescence reading was lower than that of the respective non-transduced well.

### 4.7. Statistical Analysis

All data were analyzed using IBM SPSS version 27 (IBM Corp., Armonk, NY, USA) and presented as means ± standard error of the mean (SEM). Outliers (defined as data points outside the range of mean ± 2.5× standard deviation of the data set) were removed prior to analysis. If required, data were log transformed, and if this did not achieve a normal distribution and homogeneity of variance, then non-parametric analyses were employed. For Western blotting data, a one-way ANOVA followed by post-hoc testing with Duncan’s multiple range test was used for homogenous data sets. Duncan’s multiple range test provides a robust comparison among all groups, including delineation of concentration dependent effects, as well as comparison between EPA and DHA treatments. For non-homogenous or non-normally distributed data sets, a Kruskal–Wallis test was employed followed by a least significant differences (LSD) test for pair-wise mean comparison. For sACE2 ELISA results, a paired Student’s *t*-test was employed to determine whether DHA and/or ALA had an effect on sACE2 levels after 28 days of supplementation, and to assess any possible carry-over effects of supplementation by comparing the day 0 results from both supplementation phases. A Mann–Whitney test and Student’s *t*-test (unpaired) were used to compare sACE2 between the healthy and obese groups, and between males and females within the healthy group, respectively, at baseline of the first phase. Spearman’s correlation test was used to determine associations between participant characteristics and sACE2 levels at baseline. For the pseudo SARS-CoV-2 assay, a Kruskal–Wallis test followed by LSD test was employed. Statistical significance was set at *p* < 0.05. Data graphs were plotted using GraphPad Prism Version 9.0 (GraphPad software, San Diego, CA, USA).

## 5. Conclusions

Our data demonstrate, for the first time, that the LCn3-PUFA, EPA and DHA, decrease both ACE1 and ACE2 protein levels in human EA.hy926 endothelial and HEK293 cells, and in key COVID-19-sensitive tissues of obese rats. This reduction in both non-glycosylated and glycosylated forms of ACE2 may be one of the reasons why DHA suppresses SARS-CoV-2 pseudovirus entry into cells. Furthermore, the concomitant reduction in both ACE1 and ACE2 avoids the potential detrimental effects of decreasing only ACE2. These results provide a novel perspective on how n3-PUFAs can ameliorate SARS-CoV-2 infectivity by reducing the cellular protein levels of ACE2, and interfering with the interaction of spike proteins with ACE2, whereas, to date, the emphasis has been on the well-recognized anti-inflammatory and anti-oxidative effects of n3-PUFAs [19,20]. Overall, given these positive indications, further investigation of the prophylactic potential of LCn3-PUFAs on SARS-CoV-2 infection is warranted, particularly to determine its potential utility in reducing susceptibility to infection and managing the complications associated with long COVID.

## Figures and Tables

**Figure 1 ijms-23-13825-f001:**
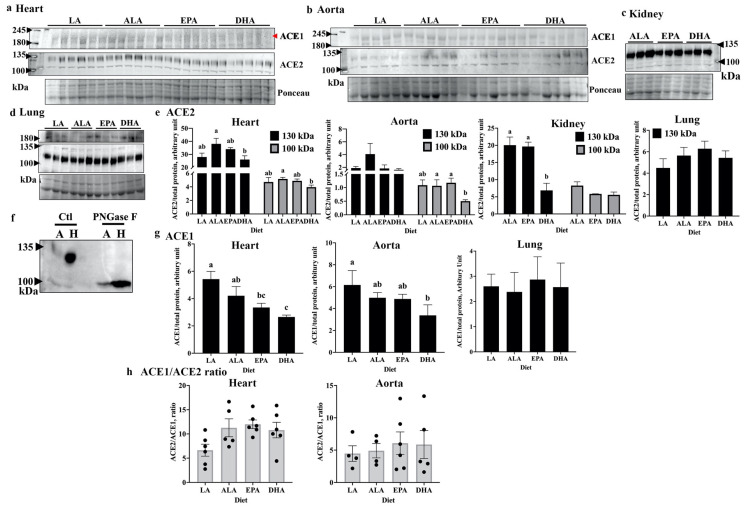
DHA reduces both ACE1 and ACE2 levels in key rat tissues. Western blotting was used to measure ACE1 and ACE2 levels relative to total protein load, as measured by Ponceau S in (**a**) the heart, (**b**) aorta, (**c**) kidney, and (**d**) lung from rats with four different diet interventions: ALA, EPA, DHA, and LA. ACE2 band intensities for the respective tissues were quantified and are graphically presented in panel (**e**). Non-glycosylated ACE2 is predicted to be 93 kDa in size (100 kDa band in the blots, grey bars), whereas the 130 kDa band (black bars) represents *N*-glycosylated ACE2. Recognition of the *N*-glycosylated form by the antibody was validated by PNGase F treatment (**f**) of the aorta (=A) and heart (=H) samples. Ctl: control samples without PNGase F treatment. ACE1 band intensities from the respective Western blots for heart (panel (**a**)), aorta (panel (**b**)), and lung (panel (**d**)) were quantified and are graphically presented in panel (**g**). The relative expression of ACE2 (sum of 100 and 130 kDa ACE2 bands in panel (**e**)) was divided by that of ACE1 to obtain the ACE1/ACE2 ratio and is graphically presented in panel (**h**). Data are presented as mean ± SEM, *n* = 5 or 6/group, except for lung where *n* = 3/group. In the graphs, bars not sharing a common letter are significantly different (*p* < 0.05) based on Duncan’s multiple range or LSD post-hoc tests.

**Figure 2 ijms-23-13825-f002:**
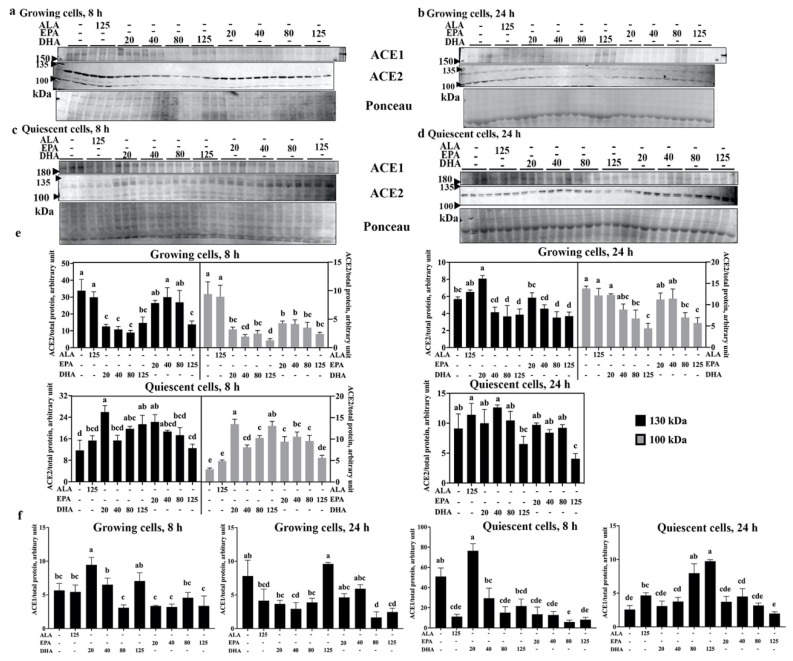
DHA differentially modulates ACE1 and ACE2 levels in growing and quiescent EA.hy926 endothelial cells. Western blotting was used to compare ACE1 and ACE2 relative to total protein load, as measured by Ponceau S in growing cells treated with ALA, EPA, or DHA at the indicated concentrations (µM) for (**a**) 8 and (**b**) 24 h, as well as in quiescent cells treated with n3-PUFA for (**c**) 8 and (**d**) 24 h. The band intensities of ACE2 and ACE1 were quantified and are graphically presented in panels (**e**,**f**), respectively; the 100 kDa band (grey bars) represents non-glycosylated ACE2, whereas the 130 kDa band (black bars) corresponds to *N*-glycosylated ACE2. Data are presented as mean ± SEM, *n* = 3; bars not sharing a common letter in the graphs are significantly different (*p* < 0.05) based on Duncan’s multiple range or LSD post-hoc tests.

**Figure 3 ijms-23-13825-f003:**
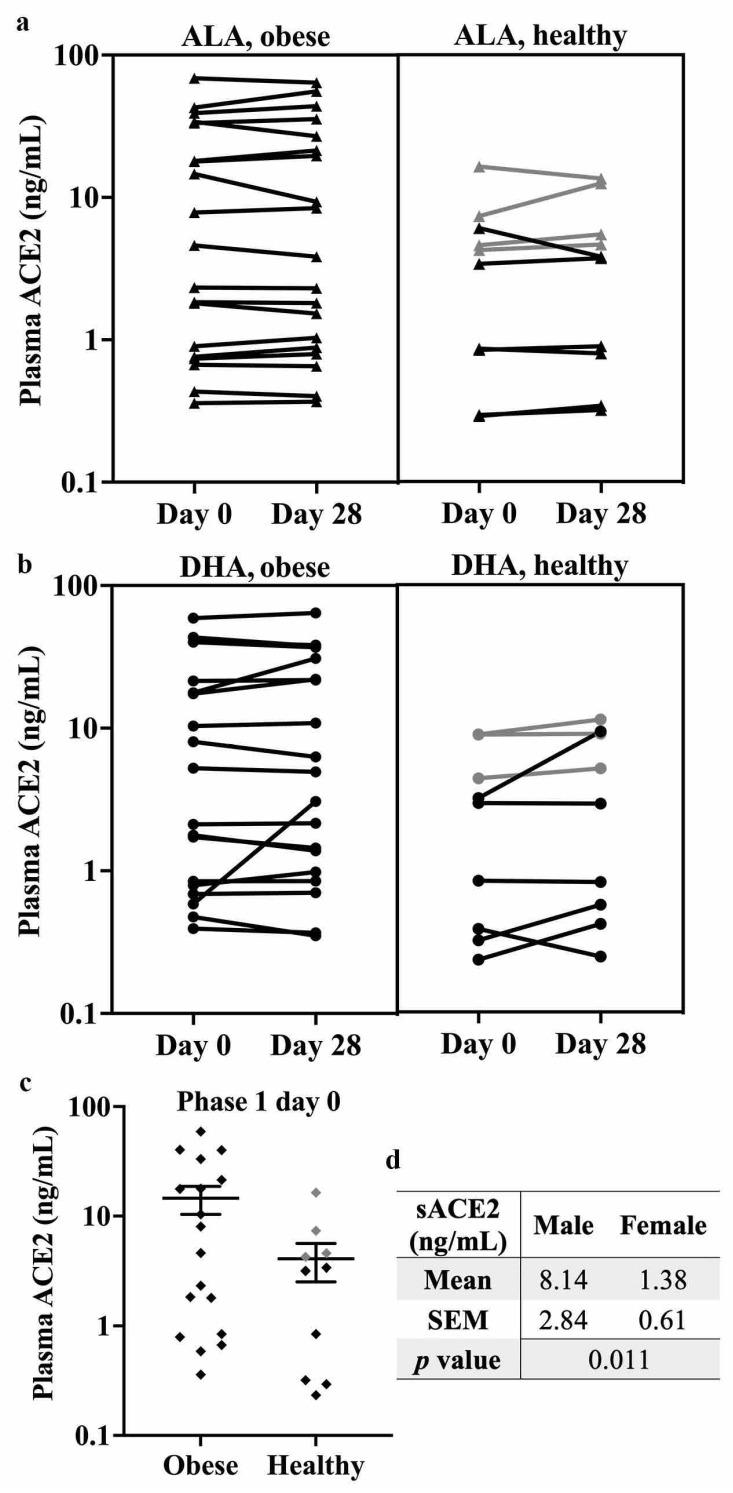
Circulating sACE2 was not affected by n3-PUFA supplementation. ELISA-quantified plasma levels of sACE2 in human subjects before and after receiving 4-week supplementation of either (**a**) 4 g ALA/d or (**b**) 4 g DHA/d in a randomized cross-over design with 4-to-6-week washout between supplementation phases. No significant effect of supplementation was detected (11.55 ± 3.11 vs. 11.86 ± 3.22, before and after ALA supplementation, respectively; 10.80 ± 3.00 vs. 11.59 ± 3.05, before and after DHA supplementation, respectively). (**c**) Baseline (Phase 1, Day 0) sACE2 levels from obese (BMI > 30 kg/m^2^) and healthy subgroups (BMI between 18 and 28 kg/m^2^; 14.54 ± 4.17 vs. 4.08 ± 1.56, respectively) with no statistically significant difference detected between the 2 groups (*p* = 0.109). (**d**) Sex differences of baseline sACE2 in the healthy subgroup. Individual data points were plotted; grey data points are male subjects and black ones are female subjects. Overall, *n* = 29 participants for both (**a**,**b**), *n* = 18 for obese and *n* = 10 for healthy in (**c**), and *n* = 4 for males and *n* = 6 for females in (**d**).

**Figure 4 ijms-23-13825-f004:**
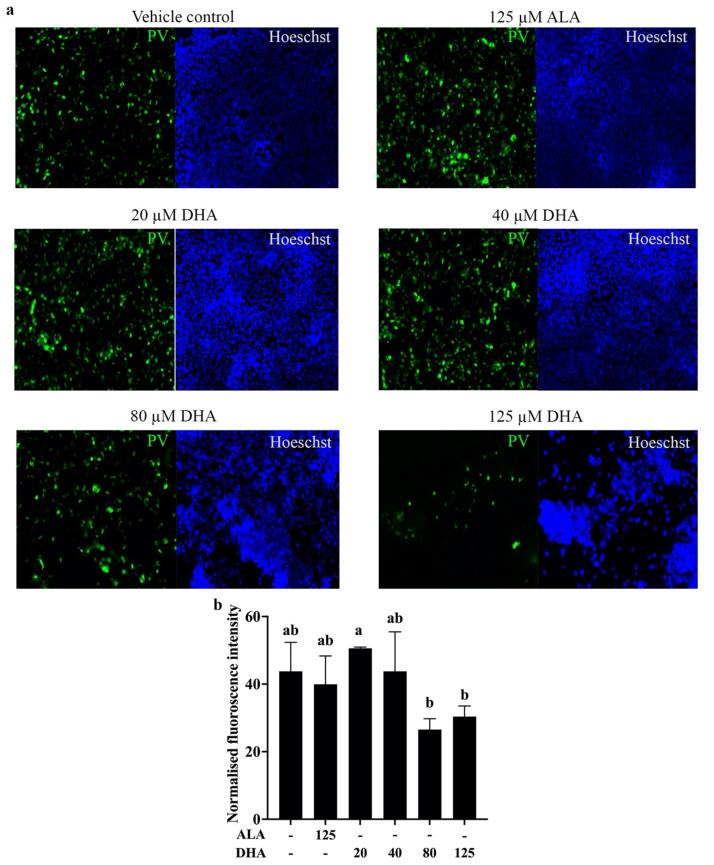
DHA reduced SARS-CoV-2 pseudovirus entry into HEK 293 cells. HEK 293 cells were treated with ALA or DHA at the indicated concentrations (µM) for 16 h before transduction for 24 h with baculovirus pseudotyped with original SARS-CoV-2 spike protein (PV) to deliver a green fluorescent reporter into the cell nuclei. Before imaging, the cell nuclei were stained with Hoechst 33342 as a control for live cell density: (**a**) Representative fluorescent images of the treated cells. Brightness and contrast were adjusted for visualization purposes to ensure the same level was maintained across the images. (**b**) Green fluorescent intensity readings of treated cells normalized to that of the Hoechst stain of live cells. Data are presented as mean ± SEM, with *n* = 3. In the graphs, bars not sharing a common letter are significantly different (*p* < 0.05), based on an LSD post-hoc test.

**Figure 5 ijms-23-13825-f005:**
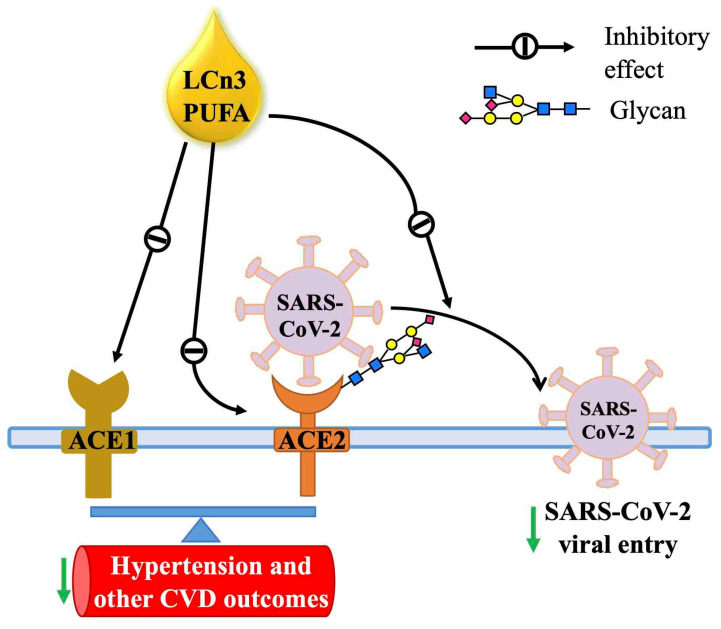
Schematic of proposed mechanism of action. LCn3-PUFAs, EPA and DHA, reduce ACE2 protein levels. These n3-PUFAs may also decrease ACE2 glycosylation. These mechanisms may explain how DHA blocks SARS-CoV-2 pseudovirus entry into HEK293 cells. LCn3-PUFAs also downregulate ACE1 protein levels, thus maintaining the balance between ACE1 and ACE2, and diminishing the risk of adverse CVD outcomes. This figure was prepared by S. Huang using Microsoft PowerPoint software version 16.16.27.

**Table 1 ijms-23-13825-t001:** Correlations between sACE2 levels and baseline health parameters of the participants.

Overall	R_S_	*p* Value
BMI vs. sACE2	0.361	0.064
SBP vs. sACE2	0.402	0.038
HDL-C vs. sACE2	−0.503	0.006
Glc vs. sACE2	0.471	0.049
**Obese subgroup ^1^**	**R_s_**	***p* value**
TC vs. sACE2	−0.439	0.069
HDL-C vs. sACE2	−0.393	0.096
Glc vs. sACE2	0.403	0.087
**Healthy subgroup ^2^**	**R_s_**	***p* value**
SBP vs. sACE2	0.669	0.049
HDL-C vs. sACE2	−0.693	0.026

^1^ Referring to participants in the NCT03583281 study [33] with a BMI > 30 kg/m^2^, *n* = 19. ^2^ Referring to participants in the NCT02317588 study [32] with a BMI between 18 and 28 kg/m^2^, *n* = 10.

## Data Availability

The study data are available upon reasonable request.

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
