# Peer review of "Long Chain N3-PUFA Decreases ACE2 Protein Levels and Prevents SARS-CoV-2 Cell Entry"

_ijms, 2022, doi:10.3390/ijms232213825_

Round 1

Reviewer 1 Report

Reviewer comment: This manuscript by Huang et al investigates omega-3 polyunsaturated fatty acids (n3-PUFAs) supplements as a prophylactic and/or therapeutic potential against

COVID-19 infection. The data provide insight for COVID-19 treatment. However, here are some issues that need to be addressed for this manuscript to be consider for publishing.

1.     In figure 1: Thus, ACE2 level did not decrease in lung, the ACE1 level in lung should still be presented.

2.     Discuss potential possibility why ACE2 level decreased in Heart Aorta and Kidney but not in lung. The author should consider performing the supplements using human transgenic human ACE2 mice, which could have a better observation of the ACE2/ACE1 level changes after treatment with n3-PUFAs supplements

3.     Fig 2 e and f, identify the range of p-value in figure legend.

4.     In 2.4 section: Please consider using ACE2-HEK293 cells, which are commercial available, to perform the DHA treatment and investigate the effect of ACE2/ACE1 level. ACE2/HEK293 cells have higher ACE2 level compared to HEK293 cells.

5.     In the material and methods, please provide CatLog number for material that was used in the experiments.

6.     Please update information for the following sections: Funding, Institutional Review Board Statement and Informed Consent Statement

7.     Line 352-353, there is a typo, please fix it:…..COVID-19 outcomes,5 and sACE2 was found to be 352 higher in obese individuals…”

Author Response

We would love to thank the reviewer for thorough assessment of the manuscript. You have provided a few great and relevant ideas for further investigation of the topic. However, due to the relative short time provided for this revision, it was not possible to add new experiments. Nevertheless, we have used the study samples to generate additional results as requested by the reviewer. We hope that the modifications made will be satisfactory. Please see the attachment for the detailed summary of these modifications.

Response to comments of Reviewer 1:

  1. In figure 1: Thus, ACE2 level did not decrease in lung, the ACE1 level in lung should still be presented.

As requested, we have prepared a Western blot of ACE1 in lungs using stored samples from our study and added the results as a new panel to Fig 1.

  1. Discuss potential possibility why ACE2 level decreased in Heart Aorta and Kidney but not in lung. The author should consider performing the supplements using human transgenic human ACE2 mice, which could have a better observation of the ACE2/ACE1 level changes after treatment with n3-PUFAs supplements

We thank the reviewer for suggesting the human ACE2 mouse model, and we will consider it for future experiments. With respect to your request that we discussed why ACE2 did not decrease in lung versus heart, aorta and kidney at line 459.

  1. Fig 2 e and f, identify the range of p-value in figure legend.

The p value for these figure panels is p <0.05. The post hoc test, Duncan’s multiple range test, enables comparison of all groups with each other (and thus identifies significant differences between different concentrations) with a significance value set at p <0.05.

  1. In 2.4 section: Please consider using ACE2-HEK293 cells, which are commercial available, to perform the DHA treatment and investigate the effect of ACE2/ACE1 level. ACE2/HEK293 cells have higher ACE2 level compared to HEK293 cells.

We were unaware that these cells were available and thank the reviewer for this information. Since our current results already found a statistically significant reduction in the pseudovirus entry, and our revision timeline is short, the suggested experiment has not been included. However, these cells should prove useful in future studies.

  1. In the material and methods, please provide CatLog number for material that was used in the experiments.

Catalogue numbers for main materials have been specified in the text as requested.

  1. Please update information for the following sections: Funding, Institutional Review Board Statementand Informed Consent Statement 

This has been completed.

  1. Line 352-353, there is a typo, please fix it:…..COVID-19 outcomes,5 and sACE2 was found to be 352 higher in obese individuals…”

The typo has been fixed to incorporate the in-text citation properly.

Reviewer 2 Report

In this article, the authors evaluate the role of omega-3 polyunsaturated fatty acids (n3-PUFA) in modulating the levels of ACE2 in rat tissues affected during COVID-19 disease, as well as in human cell lines. In addition, they analyze the levels of ACE1 and determine the ratio of these two enzymes, important for cardiovascular health. They found that some PUFA indeed decreases both ACE2 and ACE1 protein levels from relevant rat tissues and human endothelial cell lines. In addition, the authors showed that DHA pretreatment of HEK293 cells reduces the entry of SARS-CoV-2 pseudoviral particles. 

The hypothesis that n3-PUFA, especially DHA, might be a good therapeutic in the context of the COVID-19 pandemic is discussed in several reviews as plausible. However, not much data is available. Here the authors provide new evidence, yet, several points should be addressed before publication.

Major concerns:

Figure 1. A control using rats under a regular diet is missing and should be included. This is extremely relevant to judge the basal levels of ACE2 protein in the different tissues. Without this control, only differences among treatments can be observed, but no conclusion can be made regarding the effect of DHA itself, up or down-regulating ACE2 levels or having no effect. 

This is also extremely important for the consideration of the ACE2/ACE1 ratio. With the current data, you don´t know the basal ratio under a regular diet.

Another point here is how you explain that in one tissue, only the glycosylated ACE2 is downregulated, in another, the non-glycosylated version, and in others, both are.

Figure S1. Why did the authors decide to place the ACE2/ACE1 ratio as a supplement figure? In your argumentation line, this ratio is important for cardiovascular health, and it has not been extensively studied before.

Figure 2. An exact code of the letters on top of the bars is missing. Why did the authors change the system, and they do not indicate significance with a “*” symbol? If the graphs get too crowded with the * system, the significance of each treatment can be compared only to the control, and eventually, all the comparisons are included in the supplements. It is difficult to read the graphs and evaluate their significance in the current format.

For ACE2, there is a clear effect in growing cells for DHA and 125 uM of EPA. However, this is reduced at 24 hs. Which implications does this have regarding the mechanism and future treatments?

In the ACE2 quantification of part e) for quiescent cells at 8 hs, it seems that, upon EPA treatment, the levels of ACE2 increased. The same is observed at 24 hs. However, this is not reflected in the quantification. How is this explained?

Results in part f) are confusing, and no clear explanation is given as to why ACE2 levels are affected only in growing cells but for ACE1 in quiescent as well. 

What would be the mechanism behind PUFA action, considering that the ACE2 levels are diminished only in growing cells and not in the quiescent ones?

Line 230. Direct infection of the kidney by SARS-CoV-2 has to be discussed and references, including evidence, should be included. References 3 and 4 do not provide that. 

Line 235. HEK293 cells are poorly permissive to SARS-CoV-2 infection and, therefore, they are typically used overexpressing ACE2 and/or TMPRSS2. Authors should test their hypothesis in some of the cell lines widely used for SARS-CoV-2 infection, such as Caco-2 or Calu-3 cells. If a kidney cell line is needed, Human kidney cell line HK-2, described in reference 27, can be used.

Figure 4. The total number of cells (Hoeschst+) and the number of green-fluorescent cells should be quantified. Instead of fluorescence intensity, it is possible to plot “pseudo-infection” rates. 

Another point is that the total number of cells after treatment should be considered to evaluate a possible toxic effect of the DHA. 

Line 271. How does DHA reduce ACE2 protein levels in the cell? Which is the mechanism? What is the evidence to suggest that DHA might reduce ACE2 glycosylation?

A deep discussion regarding how DHA could mediate the reduction of the ACE2 protein level is missing. Several papers refer to possible mechanisms.

- Aryan, H., Saxena, A. & Tiwari, A. Correlation between bioactive lipids and novel coronavirus: constructive role of biolipids in curbing infectivity by enveloped viruses, centralizing on EPA and DHA. Syst Microbiol and Biomanuf 1, 186–192 (2021). https://doi.org/10.1007/s43393-020-00019-3

- Vivar-Sierra A, Araiza-Macías MJ, Hernández-Contreras JP, Vergara-Castañeda A, Ramírez-Vélez G, Pinto-Almazán R, Salazar JR, Loza-Mejía MA. In Silico Study of Polyunsaturated Fatty Acids as Potential SARS-CoV-2 Spike Protein Closed Conformation Stabilizers: Epidemiological and Computational Approaches. Molecules. 2021 Jan 29;26(3):711. doi: 10.3390/molecules26030711. PMID: 33573088; PMCID: PMC7866518.

- Wassall SR, Leng X, Canner SW, Pennington ER, Kinnun JJ, Cavazos AT, Dadoo S, Johnson D, Heberle FA, Katsaras J, Shaikh SR. Docosahexaenoic acid regulates the formation of lipid rafts: A unified view from experiment and simulation. Biochim Biophys Acta Biomembr. 2018 Oct;1860(10):1985-1993. doi: 10.1016/j.bbamem.2018.04.016. Epub 2018 May 3. PMID: 29730243; PMCID: PMC6218320.

- Weill P, Plissonneau C, Legrand P, Rioux V, Thibault R. May omega-3 fatty acid dietary supplementation help reduce severe complications in Covid-19 patients? Biochimie. 2020 Dec;179:275-280. doi: 10.1016/j.biochi.2020.09.003. Epub 2020 Sep 10. PMID: 32920170; PMCID: PMC7481803.

Line 324. Goc et al., 2021 showed that some PUFA interfere with the ACE-S interaction and that PUFA inhibits the activity of both TMPRSS2 and cathepsin L. This is not something to attribute to their system, this might be part of the mechanism behind your own observations of diminished pseudovirus entry.

Line 468. Do you show data regarding PUFA interfering with ACE2-S interaction?

Concerns:

Line 26. Novel application: many reviews and research articles suggest the idea of using n3-PUFA supplements as a prophylactic or therapeutic against SARS-CoV-2.

Line 46. The family is Coronaviridae. ACE2 is a receptor used only by a subgroup of viruses.

Line 60. One sentence should be included to introduce AngII production (How ACE1/ACE2 system works).

Line 62. “… this imbalance plays a major role in the...” how?

Line 93. Do the authors mean that those tissues are directly infected by SARS-CoV-2 or that they are affected during the disease? If it is the first case, reference 11 is not adequate to support direct infection of the tissues mentioned. If it is the second, it should be clarified in the sentence.

Figure 1. Please add the name of the tissues in part a-d. In e) add ACE2 and f) ACE1 and the name of tissues.

Line 125. “Phenotype” instead of “phenotypes”, otherwise which phenotypes?

Figure 2. Please add in the figure  “growing” and “quiescent” as titles, as well as 8 and 24 hs.

Line 167. I will not refer only to the 100 kDa band as the native protein. In fact, the 130 kDa, the glycosylated version of the protein is native as well.

 Line 242. Is recombinant protein going inside the cell or a full pseudoviral particle?

Line 264. Yes, but not always the two of them in the same tissue.

Line 266. Is this true for every case? You show the ratio only for the rat tissues.

Line 285. Delete “supported by evidence”. Change for “highlighted by the fact that” or similar.

Line 291. “The downregulation of ACE2 is a promising therapeutic strategy”.

Line 325. What do you mean by native here?

Line 342. What do you mean by higher sACE2? More concentration in blood?

Line 350. “likely explains” for “may be one reason why” or similar.

Line 399. Should ACE1 be ACE2?

Line 400. Reference 48 there is most likely 47. Where should reference 48 be placed?

Line 420. The functioning of the pseudovirus assay should be better explained.

Line 464. I do not agree that glycosylated ACE2 is not a native form.

Figure S2. As well as in the other figures, please use the * to denote significance. Again, native is not the proper term. Glycosylated and non-glycosylated can be used instead.

References. This reference should be incorporated and discussed.

Chiang EI, Syu JN, Hung HC, Rodriguez RL, Wang WJ, Chiang ER, Chiu SC, Chao CY, Tang FY. N-3 polyunsaturated fatty acids block the trimethylamine-N-oxide- ACE2- TMPRSS2 cascade to inhibit the infection of human endothelial progenitor cells by SARS-CoV-2. J Nutr Biochem. 2022 Jul 8;109:109102. doi: 10.1016/j.jnutbio.2022.109102. Epub ahead of print. PMID: 35817244; PMCID: PMC9264727.

Discussion. What about the effect of PUFA and, DHA in particular, on other viruses using ACE2 or not?

Minor concerns:

Line 18. (EPA) is missing.

Line 24. SARS-CoV-2.

Line 80. “Population” instead of “populace”.

Line 87. Add “in rats” or similar.

Line 116. Delete the space between “the” and “respective”.

Line 342. Delete “,” after references.

Line 352. Delete 5.

Line 361. Add “and” after [2,6,45].

Line 362. Delete “and”.

Line 374. Meaning of fa/fa rats.

Line 464. “Why” instead of “that”.

Author Response

We would love to thank the reviewer for such a comprehensive assessment of our manuscript. You have provided quite a number of excellent points for improving the manuscript, and we hope that the modifications made will be satisfactory. Please see the attachment for a detailed summary of these modifications, and the manuscript file indicated where the changes have been made with track changes.

Response to comments of Reviewer 2:

Major concerns:

  1. Figure 1. A control using rats under a regular diet is missing and should be included. This is extremely relevant to judge the basal levels of ACE2 protein in the different tissues. Without this control, only differences among treatments can be observed, but no conclusion can be made regarding the effect of DHA itself, up or down-regulating ACE2 levels or having no effect. This is also extremely important for the consideration of the ACE2/ACE1 ratio. With the current data, you don´t know the basal ratio under a regular diet.

In this study with semi-purified diets, the LA diet served as the control diet; the higher n6-PUFA composition of this diet is representative of the fatty acid composition of laboratory chow. Although this diet could be labelled as the control diet, it is more informative to identify this diet as the LA diet for readers interested in the fatty acid compositions of the diets.

  1. Another point here is how you explain that in one tissue, only the glycosylated ACE2 is downregulated, in another, the non-glycosylated version, and in others, both are.

The effect of DHA on ACE2 glycosylation is a novel observation. As indicated in the text, only one previous publication [Stupin et al, 2021 – ref 37] has reported an effect of n-3 PUFA on N-glycosylation, and this publication described the effect only on circulating plasma proteins. Thus, our study is the first to show an effect of DHA on cell surface proteins of multiple tissues. However, to address the reviewer’s comment, it is necessary to know much more about ACE2 and its N-glycosylation. Unfortunately, this information is very limited. See Rowland and Brandariz-Nuñez [2021] for a review of this material. Specifically, more is required regarding the number and location of N-glycosylation sites in relation to different tissues. It is known that 7 N-glycosylation sites exist for ACE2, however, there is significant heterogeneity of the glycans and the composition at each site remains uncharacterized [Gong et al., 2021, Shajahan et al., 2021]. Furthermore, the structures of the glycans may be tissue and/or cell type dependent [ref 36]. Note, O-glycosylation may also be present and this type of glycan could be resistant to the effects of DHA, thus allowing higher molecular mass glycosylated ACE2 to still be present even if N-glycosylation has been blocked. At this time, we cannot differentiate between these modifications and thus it is not possible to address the reviewer’s comment further. Nonetheless, it is possible to say that unglycosylated ACE2 is likely located within the endoplasmic reticulum while the glycosylated form is found on the cell surface [Rowland and Brandariz-Nuñez, 2021]. Thus, preventing or removing glycosylation likely makes ACE2 inaccessible to SARS-CoV viruses. At the same time, unglycosylated ACE2 can accumulate in the ER [Rowland and Brandariz-Nuñez, 2021], which may explain the increase in total ACE2 seen in certain situations. As indicated in several publications, treatments that affect glycosylation may be useful treatments against Covid-19 infection [Bertoldi et al., 2022, Tripathi et al., 2022]. While we cannot add more as requested by the reviewer, we do hope that the observation we have made leads to greater interest in defining the glycosylation specifics of ACE2.  On one hand, it is too early to speculate further without the requisite mechanistic studies.

- Rowland R, Brandariz-Nuñez A. Analysis of the Role of N-Linked Glycosylation in Cell Surface Expression, Function, and Binding Properties of SARS-CoV-2 Receptor ACE2. Microbiol Spectr. 2021 Oct 31;9(2):e0119921. doi: 10.1128/Spectrum.01199-21. Epub 2021 Sep 8. PMID: 34494876; PMCID: PMC8557876.

- Gong, Y., Qin, S., Dai, L. et al. The glycosylation in SARS-CoV-2 and its receptor ACE2. Sig Transduct Target Ther 6, 396 (2021). https://doi.org/10.1038/s41392-021-00809-8

- Shajahan, A., Pepi, L.E., Rouhani, D.S. et al. Glycosylation of SARS-CoV-2: structural and functional insights. Anal Bioanal Chem 413, 7179–7193 (2021). https://doi.org/10.1007/s00216-021-03499-x

- Tripathi N, Goel B, Bhardwaj N, Vishwakarma RA, Jain SK. Exploring the Potential of Chemical Inhibitors for Targeting Post-translational Glycosylation of Coronavirus (SARS-CoV-2). ACS Omega. 2022 Jul 28;7(31):27038-27051. doi: 10.1021/acsomega.2c02345. PMID: 35937682; PMCID: PMC9344791.

-Bertoldi G, Ravarotto V, Sgarabotto L, Davis PA, Gobbi L, Calò LA. Impaired ACE2 glycosylation and protease activity lowers COVID-19 susceptibility in Gitelman's and Bartter's syndromes. J Intern Med. 2022 Apr;291(4):522-524. doi: 10.1111/joim.13426. Epub 2021 Dec 16. PMID: 34875124; PMCID: PMC9414342.

  1. Figure S1. Why did the authors decide to place the ACE2/ACE1 ratio as a supplement figure? In your argumentation line, this ratio is important for cardiovascular health, and it has not been extensively studied before.

Yes, the ratio is important in cardiovascular health and other related diseases such as long COVID. But few have deliberately made such a comparison. However, based on your comment plus finding the following “Finally, the strict homeostatic balance of ACE/ACE2 activities suggests transcriptional co-regulation of both proteins.” [Saponaro et al., 2020], we have incorporated FigS1 as a panel in Fig 1 to ensure this point is emphasized better.

- Saponaro F, Rutigliano G, Sestito S, Bandini L, Storti B, Bizzarri R, Zucchi R. ACE2 in the Era of SARS-CoV-2: Controversies and Novel Perspectives. Front Mol Biosci. 2020 Sep 30;7:588618. doi: 10.3389/fmolb.2020.588618. PMID: 33195436; PMCID: PMC7556165.

  1. Figure 2. An exact code of the letters on top of the bars is missing. Why did the authors change the system, and they do not indicate significance with a “*” symbol? If the graphs get too crowded with the * system, the significance of each treatment can be compared only to the control, and eventually, all the comparisons are included in the supplements. It is difficult to read the graphs and evaluate their significance in the current format.

Duncan’s multiple range test as the post hoc provides a robust comparison among all groups with each other and the letter labelling system for significance is the most efficient for reporting statistical significance of all comparisons. By using this labelling system instead of the * system, statistical results for concentration dependent effects as well as the comparison between EPA and DHA can be shown in the figure, instead of just comparison to the control. To have consistency throughout the manuscript, we have modified Fig 1 so that Fig1, 2, and S2 (now S1) all follow the letter system.

  1. For ACE2, there is a clear effect in growing cells for DHA and 125 uM of EPA. However, this is reduced at 24 hs. Which implications does this have regarding the mechanism and future treatments?

The reduced effect after prolonged PUFA exposure in cell culture might be due to potential cytotoxicity especially at higher concentrations. Although transient plasma PUFA concentrations can reach much higher than 125 µM [ref 39], such concentrations are not maintained for 24 h. Unfortunately, in culture it is not possible to mimic the variation that might be seen over 24 hr in vivo. As mentioned in the Discussion (last paragraph), our focus was on comparing “chronic effects of low dose supplement” vs “higher dose treatment”.

  1. In the ACE2 quantification of part e) for quiescent cells at 8 hs, it seems that, upon EPA treatment, the levels of ACE2 increased. The same is observed at 24 hs. However, this is not reflected in the quantification. How is this explained?

There was no significant change among the biological triplicates after analysis of the data. For this reason, it is not possible to say anything even though visual inspection suggests there is a difference. It is likely variability in the data may be the explanation.

  1. Results in part f) are confusing, and no clear explanation is given as to why ACE2 levels are affected only in growing cells but for ACE1 in quiescent as well. 

This is a novel observation that we are also seeking to explain. Currently, no available literature dealt with these growth-state specific effects on ACE2 and ACE1. Also, this question is not relevant to the theme of the paper. More detailed mechanistic studies are yet to come.

  1. What would be the mechanism behind PUFA action, considering that the ACE2 levels are diminished only in growing cells and not in the quiescent ones?

DHA also functions as a signaling molecule and the receptors and signaling pathways PUFAs affect in the two growth states may be different. For example, EPA and DHA have different affinity to the same receptors such as FFAR4. We have incorporated this concept into the last paragraph of the Discussion.

9a. Line 230. Direct infection of the kidney by SARS-CoV-2 has to be discussed and references, including evidence, should be included. References 3 and 4 do not provide that. 

9b. Line 235. HEK293 cells are poorly permissive to SARS-CoV-2 infection and, therefore, they are typically used overexpressing ACE2 and/or TMPRSS2. Authors should test their hypothesis in some of the cell lines widely used for SARS-CoV-2 infection, such as Caco-2 or Calu-3 cells. If a kidney cell line is needed, Human kidney cell line HK-2, described in reference 27, can be used.

For the two points above, we were not looking at kidney infection in this study. Rather, HEK cells were used as a model system to study pseudovirus infectivity in relation to changes in ACE2 caused by n3-PUFA. Instead of using EA cells, we switched to HEK cells due to their greater degree of transfectability. The current data are sufficient to support that a) HEK cells express ACE2 and this expression can be downregulated by n3-PUFA treatment; and b) entry of the pseudovirus into the cells was suppressed by DHA treatment. For line 230, we understand that the current reference 3 and 4 might not be sufficient, so we added another reference published early this year by Jansen et al in Cell Stem Cell which shows the virus can infect the kidney directly.

  1. Figure 4. The total number of cells (Hoeschst+) and the number of green-fluorescent cells should be quantified. Instead of fluorescence intensity, it is possible to plot “pseudo-infection” rates. 

Yes, technically cell counting using software such as Image J is possible and we also wanted to do this. However, due to the high confluency of the cells, it was not possible to effectively separate the individual cells to due bleeding of the Hoeschst signal as can be seen in Fig 4a. This prevented us from accurately counting the number of cells and obtaining an unbiased value across all images and conditions. In addition, as explained in the Methods, the green fluorescent intensity, instead of just being present or absent in cells, is also an indication of infectivity. By only counting the fluorescent+ cells we would lose this information. Therefore, eventually we chose to use fluorescent intensity reading as our data output as recommended in the manufacturer’s protocol.

  1. Another point is that the total number of cells after treatment should be considered to evaluate a possible toxic effect of the DHA. 

The possible cytotoxicity of DHA has been explored by our lab previously as published in this paper:

Du Y, Taylor CG, Aukema HM, Zahradka P. Regulation of docosahexaenoic acid-induced apoptosis of confluent endothelial cells: Contributions of MAPKs and caspases. Biochim Biophys Acta Mol Cell Biol Lipids. 2021 May;1866(5):158902. doi: 10.1016/j.bbalip.2021.158902. Epub 2021 Feb 10. PMID: 33578050.

  1. Line 271. How does DHA reduce ACE2 protein levels in the cell? Which is the mechanism? What is the evidence to suggest that DHA might reduce ACE2 glycosylation?

The reviewer has made an excellent point, which we would be happy to know. This is a novel observation and nothing in the current literature suggests what the mechanism might be. That’s why it’s still under active investigation and is our next research question. In the discussion, we have also mentioned that one possible way is via transcriptional control since DHA was found to reduce ACE2 mRNA in samples [ref 21 and 35 in text].

  1. A deep discussion regarding how DHA could mediate the reduction of the ACE2 protein level is missing. Several papers refer to possible mechanisms.

- Aryan, H., Saxena, A. & Tiwari, A. Correlation between bioactive lipids and novel coronavirus: constructive role of biolipids in curbing infectivity by enveloped viruses, centralizing on EPA and DHA. Syst Microbiol and Biomanuf 1, 186–192 (2021). https://doi.org/10.1007/s43393-020-00019-3

- Vivar-Sierra A, Araiza-Macías MJ, Hernández-Contreras JP, Vergara-Castañeda A, Ramírez-Vélez G, Pinto-Almazán R, Salazar JR, Loza-Mejía MA. In Silico Study of Polyunsaturated Fatty Acids as Potential SARS-CoV-2 Spike Protein Closed Conformation Stabilizers: Epidemiological and Computational Approaches. Molecules. 2021 Jan 29;26(3):711. doi: 10.3390/molecules26030711. PMID: 33573088; PMCID: PMC7866518.

These 2 papers are focused on the effect of PUFA on the virus, whereas our focus is on the effects of PUFA on the cell surface receptor, ACE2.

- Wassall SR, Leng X, Canner SW, Pennington ER, Kinnun JJ, Cavazos AT, Dadoo S, Johnson D, Heberle FA, Katsaras J, Shaikh SR. Docosahexaenoic acid regulates the formation of lipid rafts: A unified view from experiment and simulation. Biochim Biophys Acta Biomembr. 2018 Oct;1860(10):1985-1993. doi: 10.1016/j.bbamem.2018.04.016. Epub 2018 May 3. PMID: 29730243; PMCID: PMC6218320.

This paper describes how DHA affects cholesterol-enriched lipid raft formation that others [Li. et al., 2021] have shown affects SARS-CoV-2 infection. Although this is relevant to the possible mechanism of how DHA can reduce SARS-CoV-2 infection via ACE2, it doesn’t provide details for how DHA could reduce ACE2 protein levels, which should be a combined result of synthesis and degradation.

- Xiaowei Li, Wenhua Zhu, Meiyang Fan, Jing Zhang, Yizhao Peng, Fumeng Huang, Nan Wang, Langchong He, Lei Zhang, Rikard Holmdahl, Liesu Meng, Shemin Lu,

Dependence of SARS-CoV-2 infection on cholesterol-rich lipid raft and endosomal acidification, Computational and Structural Biotechnology Journal,2021 (19), Pages 1933-1943, https://doi.org/10.1016/j.csbj.2021.04.001

- Weill P, Plissonneau C, Legrand P, Rioux V, Thibault R. May omega-3 fatty acid dietary supplementation help reduce severe complications in Covid-19 patients? Biochimie. 2020 Dec;179:275-280. doi: 10.1016/j.biochi.2020.09.003. Epub 2020 Sep 10. PMID: 32920170; PMCID: PMC7481803.

This paper summarized 3 possible ways by which LCn3-PUFA may help with COVID-19 infection: 1), prevent viral entry by mediating lipid rafts formation as mentioned above; 2) prevent viral replication by preventing the activation of SREBP1/2; and 3), act via their well-accepted anti-inflammatory roles. Again, while this information does provide insight into the possible mechanisms by which DHA may prevent SARS-CoV-2 infection, it does not discuss how DHA could modulate ACE2 expression. At the same time, there is little linkage to ACE2, and it doesn’t inform how DHA may reduce ACE2 protein levels and/or its glycosylation. This should, as mentioned above, involve expression, post-expression modifications, and degradation. So far, apart from the few papers we discussed in this manuscript about DHA reducing ACE2 mRNA levels, we haven’t found other literature that talks about those aspects.

  1. Line 324. Goc et al., 2021 showed that some PUFA interfere with the ACE-S interaction and that PUFA inhibits the activity of both TMPRSS2 and cathepsin L. This is not something to attribute to their system, this might be part of the mechanism behind your own observations of diminished pseudovirus entry. 

We believe that our observation that ACE2 protein levels were decreased should be more fundamental to the expression and cellular signaling level, instead of just cell surface processing. As we have discussed, ACE2 mRNA levels can be affected by DHA so the change in total ACE2 protein levels should be a combined result of synthesis (expression) and degradation. At this stage, we did not study the detailed mechanism via which DHA modulates ACE2 protein levels, so we don’t want to speculate too much without evidence. Our paper provides a novel insight into the beneficial effect of DHA on reducing SARS-CoV-2 infectivity and its adverse effect on CVD partially due to imbalance of ACE2/ACE1. A mechanistic study investigating these points is forthcoming, and at this time too much speculation is not warranted.

  1. Line 468. Do you show data regarding PUFA interfering with ACE2-S interaction?

No, this was not the objective of this study.

Concerns:

  1. Line 26. Novel application: many reviews and research articles suggest the idea of using n3-PUFA supplements as a prophylactic or therapeutic against SARS-CoV-2. 

Thank you for pointing this out. The wording has been changed in the manuscript. True, there is considerable speculation on this, but our study is the first experimental evidence using both in vitro and in vivo data to show the possible feasibility of using DHA as a therapeutic.

  1. Line 46. The family is Coronaviridae. ACE2 is a receptor used only by a subgroup of viruses.

Thanks for pointing out this. We have modified the wording in the text to ensure greater accuracy.

  1. Line 60. One sentence should be included to introduce AngII production (How ACE1/ACE2 system works).

We have added this sentence into the text: “ACE1 converts angiotensin I to angiotensin II (AngII), a potent vasoconstrictor, while ACE2 hydrolyses AngII to angiotensin-(1-7), which acts as vasodilator [14]”

  1. Line 62. “… this imbalance plays a major role in the...” how?

As mentioned in reference [14], the imbalance of ACE1/ACE2 would result in increased AngII production, thus enhancing AngII/AT1 (AngII type1 receptor) signalling, which is crucial in the pathology of diseases like cardiac remodelling, chronic lung disease, etc. In addition, this also impairs the RAS system, thus creating a vicious cycle of events involving “many cell types that produce COVID-19 pathology”. We have added “due to increased AngII signalling” in the text.

  1. Line 93. Do the authors mean that those tissues are directly infected by SARS-CoV-2 or that they are affected during the disease? If it is the first case, reference 11 is not adequate to support direct infection of the tissues mentioned. If it is the second, it should be clarified in the sentence. 

Thanks for pointing out the ambiguity here. We were referring to the second case and have modified the expression in text to “..key targets affected by SARS.. infection [11].”

  1. Figure 1. Please add the name of the tissues in part a-d. In e) add ACE2 and f) ACE1 and the name of tissues. 

Have added these in the figure.

  1. Line 125. “Phenotype” instead of “phenotypes”, otherwise which phenotypes?

Here we were referring to change from healthy phenotype to dysfunctional phenotype.

  1. Figure 2. Please add in the figure  “growing” and “quiescent” as titles, as well as 8 and 24 hs. 

Modified in the figure.

  1. Line 167. I will not refer only to the 100 kDa band as the native protein. In fact, the 130 kDa, the glycosylated version of the protein is native as well. 

Thanks for this insight. We have modified the format throughout the manuscript to refer to the two forms as glycosylated and non-glycosylated.

  1. Line 242. Is recombinant protein going inside the cell or a full pseudoviral particle?

The kit provides a baculovirus system pseudotyped with the S proteins. The pseudovirus will go inside the cell but just deliver a genetically-encoded fluorescent reporter into the host cell nucleus.

  1. Line 264. Yes, but not always the two of them in the same tissue.

Yes, specifically speaking, DHA only significantly reduced non-glycosylated ACE2 in rat aorta but not the p130 form, for instance, while both forms were reduced in heart (Fig. 1) and cultured cells (Fig. 2 and S2). However, in this paragraph, we just want to generally summarise our overall findings.

  1. Line 266. Is this true for every case? You show the ratio only for the rat tissues.

It’s true in most cases except for 125 µM DHA in growing cells due to the failure of ACE1 to change compared to the reduction in ACE2 levels. However, first, the in vivo data should reveal more physiological relevant results compared to in vitro data. Second, as we mentioned in the Discussion, the proper treatment regimen should still be considered carefully, especially when translating in vitro data to in vivo application.

  1. Line 285. Delete “supported by evidence”. Change for “highlighted by the fact that” or similar.

Thanks; it has been changed in the text

  1. Line 291. “The downregulation of ACE2 is a promising therapeutic strategy”.

Thanks; it has been changed in the text.

  1. Line 325. What do you mean by native here?

Here the meaning is the ACE2 that’s expressed by the cells and tested in live cells, instead of recombinant ACE2 as in the Goc et al paper.

  1. Line 342. What do you mean by higher sACE2? More concentration in blood? 

Yes, we define sACE2 as ACE2 circulating in the plasma. Higher sACE2 means there is a higher concentration in the blood.

  1. Line 350. “likely explains” for “may be one reason why” or similar.

 Thanks; it has been changed in the text.

  1. Line 399. Should ACE1 be ACE2? 

We have double checked the expression, which is correct in that place.

  1. Line 400. Reference 48 there is most likely 47. Where should reference 48 be placed?

Ref 48 is in the correct place, but Ref 47 a few lines above was mis-labelled. Sorry for the mistake. It has been changed in the text.

  1. Line 420. The functioning of the pseudovirus assay should be better explained.

We have added a few lines in the Methods section to further explain the pseudovirus assay and hope this will help with understanding the model as well as the results.

  1. Line 464. I do not agree that glycosylated ACE2 is not a native form.

Yes, we do understand where you are coming from now and thanks so much for pointing that out. As mentioned in the point 24-line 167, we have changed all the instances to non-glycosylated instead.

  1. Figure S2. As well as in the other figures, please use the * to denote significance. Again, native is not the proper term. Glycosylated and non-glycosylated can be used instead.

Changed in the text for non-glycosylation. We have standardized how statistical significance is denoted throughout the manuscript.

  1. References. This reference should be incorporated and discussed.

Chiang EI, Syu JN, Hung HC, Rodriguez RL, Wang WJ, Chiang ER, Chiu SC, Chao CY, Tang FY. N-3 polyunsaturated fatty acids block the trimethylamine-N-oxide- ACE2- TMPRSS2 cascade to inhibit the infection of human endothelial progenitor cells by SARS-CoV-2. J Nutr Biochem. 2022 Jul 8;109:109102. doi: 10.1016/j.jnutbio.2022.109102. Epub ahead of print. PMID: 35817244; PMCID: PMC9264727.

Thanks for letting us know about this very relevant paper that was published recently. We have incorporated it into the Discussion (line 376).

  1. Discussion. What about the effect of PUFA and, DHA in particular, on other viruses using ACE2 or not?

We have added commentary about the effect of PUFA on other viruses into the Discussion (line 355), using the reference provided in the section above.

So far, ACE2 is the receptor for coronaviruses closely related to SARS-CoV. We don’t think it’s relevant to the overall theme of this paper to discuss the effect of PUFA on other viruses that do not use ACE2 as their entry point.

  1. Minor concerns:

All have been corrected in the text with track changes unless otherwise specified

Line 18. (EPA) is missing.

Line 24. SARS-CoV-2.

Line 80. “Population” instead of “populace”.

Line 87. Add “in rats” or similar.

Line 116. Delete the space between “the” and “respective”. 

Line 342. Delete “,” after references. 

Line 352. Delete 5. Instead of deleting, this is a misplaced reference. We have modified it.

Line 361. Add “and” after [2,6,45].

Line 362. Delete “and”.

Line 374. Meaning of fa/fa rats.

These are an “obese” model.

“The fa mutation is an autosomal recessive locus on chromosome 5, and homogeneity results in an improperly coded leptin receptor gene (Baron et al. 1993).”

- T.R. Nurkiewicz, J.C. Frisbee, M.A. Boegehold,6.08 - Assessment of Vascular Reactivity, Editor: Charlene A. McQueen, Comprehensive Toxicology (Second Edition), Elsevier, 2010, Pages 133-148, ISBN 9780080468846, https://doi.org/10.1016/B978-0-08-046884-6.00707-7.

Line 464. “Why” instead of “that”.

Round 2

Reviewer 2 Report

Figure 2. Did you include the references you mentioned and part of this discussion in the manuscript?

11. Ok, then mention how toxic DHA is in these cells using and which toxicity assay you used before, if you have CC50s as well, because there is a clear effect on the cells at 80 and 125 uM, considering the Hoechst staining of the cells.

Author Response

We appreciate the additional input from the reviewer following up to the previous revision. We have clarified both follow-up questions from the reviewer, and the manuscript file has also been edited accordingly. We thank the reviewer for his or her time and effort in assessing our manuscript, and hope that our answers clarifies your concern.
